# Metformin Increases Proliferative Activity and Viability of Multipotent Stromal Stem Cells Isolated from Adipose Tissue Derived from Horses with Equine Metabolic Syndrome

**DOI:** 10.3390/cells8020080

**Published:** 2019-01-22

**Authors:** Agnieszka Smieszek, Katarzyna Kornicka, Jolanta Szłapka-Kosarzewska, Peter Androvic, Lukas Valihrach, Lucie Langerova, Eva Rohlova, Mikael Kubista, Krzysztof Marycz

**Affiliations:** 1Department of Experimental Biology, The Faculty of Biology and Animal Science, University of Environmental and Life Sciences, 50375 Wroclaw, Poland; kornicka.katarzyna@gmail.com (K.K.); jolanta.szlapka@upwr.edu.pl (J.S.-K.); krzysztof.marycz@upwr.edu.pl (K.M.); 2Laboratory of Gene Expression, Institute of Biotechnology CAS, Biocev, 25250 Vestec, Czech Republic; peter.androvic@ibt.cas.cz (P.A.); lukas.valihrach@ibt.cas.cz (L.V.); eva.rohlova@ibt.cas.cz (E.R.); mikael.kubista@ibt.cas.cz or mikael.kubista@tataa.com (M.K.); 3Laboratory of Growth Regulators, Faculty of Science, Palacky University, 78371 Olomouc, Czech Republic; 4Gene Core BIOCEV, Průmyslová 595, 25250 Vestec, Czech Republic; lucie.langerova@ibt.cas.cz; 5Department of Anthropology and Human Genetics, Faculty of Science, Charles University, 12843 Prague, Czech Republic; 6TATAA Biocenter AB, 41103 Gothenburg, Sweden; 7Faculty of Veterinary Medicine, Equine Clinic-Equine Surgery, Justus-Liebig-University, 35392 Giessen, Germany

**Keywords:** adipose-derived stromal cells, equine metabolic syndrome, metformin

## Abstract

In this study, we investigated the influence of metformin (MF) on proliferation and viability of adipose-derived stromal cells isolated from horses (EqASCs). We determined the effect of metformin on cell metabolism in terms of mitochondrial metabolism and oxidative status. Our purpose was to evaluate the metformin effect on cells derived from healthy horses (EqASC_HE_) and individuals affected by equine metabolic syndrome (EqASC_EMS_). The cells were treated with 0.5 μM MF for 72 h. The proliferative activity was evaluated based on the measurement of BrdU incorporation during DNA synthesis, as well as population doubling time rate (PDT) and distribution of EqASCs in the cell cycle. The influence of metformin on EqASC viability was determined in relation to apoptosis profile, mitochondrial membrane potential, oxidative stress markers and *BAX*/*BCL-2* mRNA ratio. Further, we were interested in possibility of metformin affecting the Wnt3a signalling pathway and, thus, we determined mRNA and protein level of *WNT3A* and β-catenin. Finally, using a two-tailed RT-qPCR method, we investigated the expression of *miR-16-5p*, *miR-21-5p*, *miR-29a-3p*, *miR-140-3p* and *miR-145-5p*. Obtained results indicate pro-proliferative and anti-apoptotic effects of metformin on EqASCs. In this study, MF significantly improved proliferation of EqASCs, which manifested in increased synthesis of DNA and lowered PDT value. Additionally, metformin improved metabolism and viability of cells, which correlated with higher mitochondrial membrane potential, reduced apoptosis and increased *WNT3A*/β-catenin expression. Metformin modulates the miRNA expression differently in EqASC_HE_ and EqASC_EMS_. Metformin may be used as a preconditioning agent which stimulates proliferative activity and viability of EqASCs.

## 1. Introduction

Currently, equine metabolic syndrome (EMS) is considered as a burning issue in veterinary medicine, affecting more and more horses. By definition, EMS is related to insulin resistance (IR), insulin dysregulation and obesity, as well as hyperleptinemia and past or chronic laminitis. Until recently, it was thought that only primitive-type horses, such as ponies and cold-bloods, suffer from EMS, but recent findings suggest that also non-obese and even sport horses might be affected by EMS—mainly due to a high starch and high energy diet [1]. Although obesity was excluded as a sine qua non condition in the course of the diagnostic procedure of EMS, specific local accumulation of adipose tissue, i.e., adiposity (cresty neck) is considered a diagnostic marker. Moreover, it is thought that EMS horses that are not overweight might accumulate abdominal adipose tissue, which is not without physiological significance for the other organs, including liver. Adipose tissue produces a number of factors, including cytokines, adipokines, as well as hormones, all influencing the clinical picture of EMS horses. It was shown that adipocytes isolated from subcutaneous adipose tissue of EMS horses produce pro-inflammatory cytokines, i.e., tumour necrosis factor-alpha (TNF-α), interleukin 1 (IL-1) and interleukin 6 (IL-6), which all may lead to the development of local inflammation [2]. Our previous studies indicate that subcutaneous adipose tissue inflammation is mediated by tissue resident immune cells, including macrophages that, under EMS condition, are characterised by elevated activity. This unfavourable pro-inflammatory microenvironment of adipose tissue has an adverse effect on residing progenitor cells, i.e., adipose-derived multipotent stromal stem cells (ASCs). Equine ASCs are characterised by the presence of mesenchymal specific surface antigens, including CD73, CD90 and CD105, and lack of expression of hematopoietic markers, i.e., CD45 [3,4,5]. Additionally, this population of cells is endowed with self-renewal properties regulated by the expression of *OCT4* (octamer binding transcription factor-4), *SOX2* (sex-determining region Y-box 2) and homeobox protein Nanog [6]. Furthermore, it was shown that ASCs possess immunomodulatory properties and secrete anti-inflammatory cytokines, such as IL-4 and IL-13. The increased proliferative activity and immunomodulatory properties of ASC, along with low immunogenicity, makes them promising a therapeutic tool for the treatment of various musculoskeletal diseases in horses [7]. ASCs, in general, are also characterised by unique ability for multilineage differentiation, including osteogenic, adipogenic and chondrogenic, which is crucial for their clinical use. Our own previous clinical research showed a positive effect of ASCs in horses with particular musculoskeletal system disorders [8,9]. In general, the pro-regenerative properties of ASCs are explained by their autocrine and paracrine activity [10]. For example, it was shown that application of ASCs in injured Achilles tendons is more efficient than the application of growth differentiation factor 5 (GDF-5). The transplantation of ASCs increased the expression of several genes (including *TGFβ*), which improved collagen fibre organisation and tendon biomechanics [11]. Additionally, it was demonstrated that equine ASCs are able to synthesise and secrete extracellular microvesicles (ExMVs), rich in broad range of growth factors, including bone morphogenetic protein isoform 2 (BMP-2) and vascular endothelial growth factor (VEGF) [12]. Various studies, including ours, demonstrated that regenerative potential of ASCs depends strictly on donor age or its physiological status [13,14,15,16]. Importantly, in our previous research, we have shown that ASCs derived from EMS horses (EqASC_EMS_) show impaired proliferative and metabolic activity, reduced clonogenic potential, as well as lowered expression of KI-67, a widely known proliferation marker. Furthermore, when determining the multipotency of EqASC_EMS_, we noticed that their chondrogenic and osteogenic differentiation potential had declined, which was associated with the reduced expression of transcripts such as *BMP-2*, *SOX-9*, *COL-1/2* and vimentin [5]. Moreover, in EqASC_EMS_, we have observed deterioration of mitochondrial dynamics, which is related to lowered mitochondrial metabolism and induced macroautophagy process. The results question the utility of EqASC_EMS_ in terms of autologous transplants, that are considered as well-established therapeutic strategies for the treatment of tendon and joint diseases [8,9,17,18]. Bearing in mind these facts, we see great need for the development of new preconditioning regimens to enhance the regenerative potential of EqASC_EMS_. Most recently, our group has shown that EqASC_EMS_ displayed anti-inflammatory properties and decreasing activity of TNF-α, IL-1 and IL-6 when preconditioned with a combination of 5-azatacidine and resveratrol (AZA/RES). The preconditioned cells were able to regulate and activate the anti-inflammatory response related to regulatory T lymphocytes (T_REG_) [19]. Additionally, we have shown that AZA/RES may rejuvenate EqASC_EMS_ by modulating mitochondrial dynamics and increasing their viability [20]. Our previous studies indicate that metformin and biguanide, both anti-diabetic drugs, can be considered as promising candidates in terms of improving progenitor cells’ viability and their proliferative potential. Using the ex vivo model, we showed that metformin is able to increase the proliferative activity and viability of mice ASCs (mASCs). The pro-proliferative effect of metformin towards mASCs was manifested by increased proliferation ratio, lowered population doubling time and enhanced clonogenic potential [21]. Moreover, our other studies have shown that metformin may also improve viability and stabilise the phenotype of mouse glial progenitor cells, i.e., olfactory ensheathing cells (mOECs), without influence on their proliferative status [22]. Our studies showed that increased viability of progenitor cells after metformin treatment may be associated with its antioxidant effect and improved metabolism of mitochondria [21,22]. Additionally, it was shown that metformin suppresses proinflammatory responses of adipocyte and improves the balance of brown/white adipose acting upon obesity effects [23,24,25]. Furthermore, some clinical studies showed the beneficial effect of metformin in terms of insulin resistance treatment in horses. For example, it was shown that metformin can reduce glycaemic and insulinaemic responses both in healthy horses and in horses with experimentally induced insulin resistance [26]. There is also data indicating that metformin reverses insulin resistance and decreases serum insulin concentration during the first 6 to 14 days of treatment, however, this effect diminishes by 220 days [27]. The clinical efficacy of metformin in terms of EMS treatment has not been proven, due to some questions concerning its bioavailability [28,29]. Still, being aware of pro-regenerative effects of metformin towards progenitor cells [21,22] and its pro-aging activities [30], we decided to characterise metformin influence on viability and proliferative potential of EqASC_EMS_. We determined the effect of metformin on cells morphology, apoptosis profile and mitochondrial membrane activity. We analysed the antioxidative and anti-apoptotic effect of metformin in terms of expression of several markers both on mRNA and miRNA level. We tested the expression of *BAX* and *BCL-2*, as well as *miR-16-5p*, *miR-21-5p*, *miR-29a-3p*, *miR-140-3p* and *miR-145-5p*. The specificity of miRNA measurement was assured by highly sensitive two-tailed RT-qPCR method [31]. It is well known that metformin acts through AMP-activated protein kinase (AMPK) which regulates lipid, cholesterol and glucose metabolism in various metabolic tissues, including adipose tissue, yet, it was also shown that metformin may improve cells survival through WNT/β-catenin signalling [32]. Therefore, we were also interested in whether *WNT* signalling is activated in EqASC_EMS_ after metformin treatment_._ The obtained results show promise for the potential application of metformin as a preconditioning agent, improving cellular health of adipose-derived multipotent stromal cells isolated from horses with equine metabolic syndrome (EqASC_EMS_).

## 2. Materials and Methods

### 2.1. Characterisation of Equine Multipotent Stromal Cells (EqASCs)

Cells derived from healthy horses (*n* = 6) and horses affected by metabolic syndrome (*n* = 6) were used in the study. The method to classify the animals has been detailed previously [1,2,3,4]. Subcutaneous adipose tissue collected from horses’ tail base was used for isolation of EqASCs. The procedure of tissue collection was performed with the standard surgical protocols in compliance with ethical standards and approved by the II Local Ethics Committee of Environmental and Life Sciences University (Chelmonskiego 38C, 51–630 Wroclaw, Poland; decision No. 84/2012; extension No. 84/2018). The multipotent stromal cells were isolated from the stromal vascular fraction obtained by enzymatic digestion of adipose tissue using collagenase type I. The precise protocol of EqASC isolation was previously described in detail [3,4,5,33,34]. Primary cultures of EqASCs were maintained in Dulbecco’s Modified Eagle’s Medium (DMEM) with F-12 Ham nutrient. The medium was supplemented with 10% foetal bovine serum (FBS) and 1% of antibiotic solution containing penicillin, streptomycin and amphotericin B (PSA). Constant and aseptic growth conditions were assured by maintaining the cells in CO_2_ incubator at 37 °C and 95% humidity. Cultures were passaged using trypsin solution (TrypLE Express; Life Technologies, Warsaw, Poland) after reaching 80% confluence. Complete growth medium (CGM) used for the subsequent EqASC cultures consisted of DMEM containing 4500 mg/L glucose supplemented with 10% FBS and 1% of PSA. The media were changed every two days. The cells used for experiments were passaged three times and were characterised as multipotent stromal cells based on a specific phenotype and ability to differentiate into adipocytes, chondrocytes and osteocytes [3,4,5].

### 2.2. The Experimental Cultures

The multipotent stromal cells isolated from adipose tissue derived from healthy horses (EqASC_HE_) and horses with equine metabolic syndrome (EqASC_EMS_) were inoculated in a 24-well plates. The initial inoculum was 30,000 cells per well. The cells were cultivated for 72 h in CGM containing metformin at a final concentration equal to 500 μM. The control for the experiment was EqASCs maintained in CGM without metformin.

### 2.3. The Analysis of Metformin Influence on Morphology of EqASCs and Metabolic and Proliferative Activity

The morphology of cells was evaluated using an epifluorescence microscope (EpiFM). For the analysis, cultures were fixed using 4% paraformaldehyde and stained with atto-488-labeled phalloidin (1:800) and with diamidino-2-phenylindole (DAPI; 1:1000). The observations were performed using Axio Observer A.1 microscope (Zeiss, Oberkochen, Germany). The metabolic activity of cells was monitored every 24 h using Alamar Blue assay and, additionally, population doubling time was determined accordingly to the method described previously [22,35]. The distribution of cells in the cell cycle was determined using Muse™ Cell Analyzer (Merck KGaA, Darmstadt, Germany) using the Cell Cycle Assay Kit (Merck, Warszawa, Poland). The assay was performed following manufacturer’s instructions. Each analysis was performed in triplicate.

### 2.4. The Analysis of Metformin Influence on Mitochondrial Metabolism of EqASCs

The mitochondrial membrane potential was assessed with Muse™ Cell Analyzer. After culture, cells were harvested with trypsin solution and counted with trypan blue solution using a standard protocol [36]. For the assay, 100,000 cells were stained with MitoPotential kit (Merck, Warszawa, Poland). Staining was performed according to the protocol provided by manufacturer (Merck, Warszawa, Poland). Each measurement was performed at least three times. The ultrastructural analyses of EqASC mitochondria were performed using focused ion beam microscope (FIB, Cobra, AURIGA 60, Zeiss, Oberkochen, Germany). The protocol of preparing the material for FIB imaging was previously described by Marycz et al. [5,37]. The analysis was conducted using an SE2 detector (Zeiss, Oberkochen, Germany) at 2 kV of electron beam voltage. Moreover, the influence of metformin on mitochondrial metabolism was determined based on oxidative stress factors accumulation. The supernatants after 72 h of culture were collected in order to evaluate the activity of intracellular reactive oxygen species (ROS), nitric oxide (NO) and superoxide dismutase (SOD). ROS were measured using H2DCF-DA solution, while NO activity was measured using Griess reagent kit (both reagents from Thermo Fisher Scientific, Warszawa, Poland). SOD was determined using a commercially available SOD determination kit (Sigma Aldrich, Munich, Germany).

### 2.5. The Analysis of Metformin Influence on the Viability of EqASC

The apoptosis and necrosis were quantified using Muse™ Cell Analyzer. For this purpose, 100,000 cells were stained with Muse^®^ Annexin V and Dead Cell Assay Kit (Merck, Warszawa, Poland). The procedure of staining was performed following the manufacturer’s protocol. The analysis was performed three times. Moreover, the viability of cultures was investigated using a well-established staining method [33,38] with a two-colour fluorescence live/dead assay according to the manufacturer’s instructions (Double Staining Kit: Sigma Aldrich, Munich, Germany). The cultures were analysed using an epifluorescence microscope (Axio Observer A.1; Zeiss, Oberkochen, Germany) and images were captured using a PowerShot camera (Canon, Warszawa, Poland).

### 2.6. Influence of Metformin on Endogenous Levels of WNT3A and β-Catenin

The expression of endogenous WNT3A and β-catenin was determined using Western blot technique. After harvesting, cells were lysed with ice-cold RIPA extraction buffer. The extraction buffer contained 1% of protease and phosphatase inhibitor cocktail (Sigma Aldrich, Munich, Germany). To normalise the amount of protein loaded onto the gel, the total concentration of protein in the samples was determined using the Bicinchoninic Acid Assay Kit (Sigma Aldrich, Munich, Germany). The cell extracts containing 50 μg of protein were separated using 10% sodium dodecyl sulphate-polyacrylamide gel electrophoresis (SDS-PAGE; 30 mA ~80 min) and transferred to nitrocellulose membrane at 100 V for 1 h at 4 °C in Tris/glycine buffer using the Mini Trans-Blot^®^ system (Bio-Rad, Hercules, CA, USA). After transfer, the membranes were washed with Tris/NaCl/Tween buffer (TBST) and blocked overnight at 4 °C with 5% bovine serum albumin (BSA). Membranes were then washed twice with TBST, and three times with TBS. Each rinsing lasted 5 min and was performed at room temperature under agitation (15 rpm). Next, the membranes were incubated for 2 h at room temperature with primary antibodies detecting Wnt-3a (SAB2105736), phospho-β-catenin (SAB4300630) and β-actin (A2066) that were prepared at a dilution of 1:200 in 5% of BSA in TBST. After incubation with primary antibody, the membranes were washed again as described above. After rinsing, membranes were incubated with secondary antibody conjugated with alkaline phosphatase (A9919) for 1.5 h at room temperature. All antibodies were from Sigma Aldrich (Munich, Germany). The membranes were washed and incubated with BCIP^®^/NBT-Purple Liquid Substrate (Sigma Aldrich, Munich, Germany) for 10 min. The reaction was stopped by washing the membrane with distilled water. The Western blot analysis was repeated twice. The blots were analysed using Bio-Rad ChemiDoc™ XRS system. The signals were captured from the bands and the intensity was quantified using Image Lab™ Software (Bio-Rad).

### 2.7. The Analysis of Metformin Influence on Expression of Genes Associated with Apoptosis

The cultures were washed with Hanks’ Balanced Salt solution (HBSS) and homogenised directly in culture dishes using TRI Reagent^®^ (Sigma Aldrich, Munich, Germany). The isolation of total RNA was performed accordingly to the protocol published by Chomczynski and Sacchi [39]. The quantity of RNA was measured using NanoDrop 8000 (ThermoFisher Scientific, Waltham, MA, USA).

#### 2.7.1. The Analysis of mRNA Expression

The cDNA used for the qPCR was obtained from 1 µg of RNA and was synthesised accordingly to a method described previously [40]. SensiFast SYBR & Fluorescein Kit (Bioline Reagents Ltd., London, United Kingdom) was used for the detection of specific amplicons. The total volume of PCR was 10 μL, and cDNA did not exceed 10% of the final PCR mix volume, while the concentration of the primers was 400 nM. The primer sequences have been published previously [41,42]. All primers were synthesised by Sigma Aldrich (Sigma Aldrich, Munich, Germany). The qPCR was performed applying CFX Connect Real-Time PCR Detection System (Bio-Rad Polska Sp. z.o.o., Warszawa, Poland) using a protocol described elsewhere [43]. The transcript levels were normalised to the expression of reference gene housekeeping gene, i.e., glyceraldehyde 3-phosphate dehydrogenase (*GAPDH*).

#### 2.7.2. The Analysis of miRNA Expression

The quality of RNA was tested using capillary electrophoresis (Fragment Analyser, Agilent Technologies, Inc., BioVendor, Brno, Czech Republic). Only fully intact RNA was used for further analysis. The RT was performed from 10 ng of RNA using qScript Flex cDNA Kit (QuantaBio; Beverly, MA, USA). For the reaction, target-specific primers were used, designed according to principles described previously [31]. The primers are listed in Appendix A. The final concentration of primers in each reaction was 50 nM. To monitor the technical aspects of the experiment, a mix of artificial spike-in miRNA molecules was added into each sample prior to RNA extraction and RT. The RT reaction was performed in accordance to the protocol provided by the producer. Obtained cDNA was 10-times diluted and used in subsequent qPCR reactions measuring the expression of miRNAs, reference gene (snU6) and 5 spike assays. Each reaction was performed in triplicates. The reaction mix was composed of TATAA SYBR^®^ GrandMaster^®^ Mix (TATAA), 400 nM of primers (ThermoFisher Scientific), 2.6 μL nuclease free water (ThermoFisher Scientific) and 2 μL of diluted cDNA. qPCR was performed in CFX384 instrument (Biorad, Hercules, California USA). The following temperature profile was used: 95 °C for 30 s, 45 cycles of amplification (95 °C for 5 s and 60 °C for 15 s). The specificity of products was determined based on melting curve analysis. The primer sequences used for qPCR are shown in Appendix A. RT-qPCR data were processed and analysed with GenEx software (MultiD, Sweden). Cq values were normalised to a reference gene (snU6), transformed into relative quantities (scaled to the sample having the lowest expression), and converted into log2 scale as described previously [44,45]. Significant differences between data were tested by analysis of variance (ANOVA).

## 3. Results

### 3.1. Metformin Improves Metabolic Activity and Proliferation of EqASCs

Obtained results revealed that metformin may act as an agent that increases the proliferative activity of EqASCs derived both from healthy and EMS horses (Figure 1). Microscopic evaluation of EqASC_HE_ and EqASC_EMS_ cultures showed that metformin does not affect the cells’ morphology—the cells maintain proper fibroblast-like morphotype. However, the distribution of cells and the growth pattern indicated on increased confluency of cultures treated with metformin (Figure 1a). Direct evidence of pro-proliferative activity of metformin towards EqASCs was found in shortened population doubling time (PDT). The PDT decreased after metformin treatment in ASC cultures derived from both healthy and EMS horses (Figure 1b,c). The metformin influenced the metabolic activity of cells, which was visible, in particular, in significant improvement of metabolic activity in cultures after 48 h of propagation. The increased metabolic activity of cells maintained for 72 h of culture is shown in Figure 1d,e.

The analysis of cell cycle showed that metformin treatment may change the distribution of EqASCs and induce their shift towards S-phase. Simultaneously, we observed the decrease of percentage of cells in G0/G1-phase. Moreover, the metformin significantly (*p* < 0.05) increased the number of EqASC_HE_ in G2/M-phase (Figure 2).

### 3.2. Metformin Enhances Mitochondrial Potential and Improves Oxidative Status in EqASCs

The analysis of mitochondrial membrane potential confirmed that metformin improves the metabolic activity of EqASCs isolated both from healthy and from EMS individuals (Figure 3a). The number of cells with improved mitochondrial potential increased significantly, both in EqASC_HE_ and EqASC_EMS_ cultures, after metformin treatment (Figure 3c). Simultaneously, the percentage of total depolarised cells decreased in cultures treated with metformin (Figure 3d). Nevertheless, the impairment of mitochondrial function in EqASC_EMS_ remained significant when compared to EqASC_HE_, and metformin did not reverse mitochondrial deterioration due to EMS. We did not observe significant changes of mitochondria morphology in EqASC_HE_ treated with metformin; in these cells, mitochondria had proper shape and morphology as well as visible cristae (Figure 3b). Analysis of EqASC_EMS_ ultrastructure showed that, in cultures treated with metformin, the number of mitochondria increased. Additionally, elongated mitochondria were noted (Figure 3b). These ultrastructural features, along with the lowered activity of reactive oxygen species (ROS, Figure 3e) and nitric oxide (NO, Figure 3f), may indicate that metformin may improve the elimination of dysfunctional mitochondria by autophagy, enhancing mitochondrial dynamics. Additionally, in EqASC_EMS_ cultures treated with metformin, we observed increased levels of superoxide dismutase (SOD, Figure 3g).

### 3.3. Metformin Increases Viability of EqASC Cultures

Quantification of apoptosis and necrosis by annexin V binding and propidium iodide uptake revealed that metformin exerts an anti-apoptotic effect on EqASC_EMS_ cultures (Figure 4a). The cellular viability of EqASC_EMS_ was significantly restored following metformin treatment. The percentage of early and late apoptotic cells decreased after metformin administration, not only in EqASC_EMS_ but also in EqASC_HE_ (Figure 4c,d). Additionally, the viable cells were visualised with calcein-AM staining, while dead cells were counterstained with propidium iodide. The images confirmed the previous observations that metformin increases confluency of EqASCs in cultures. Nevertheless, based on the pictures, it was difficult to ascertain the influence of metformin on cells viability, because dead cells were dimly fluorescent in analysed cultures (Figure 4b). The anti-apoptotic effect of metformin towards EqASCs was confirmed using RT-qPCR. We determined that metformin decreases mRNA expression of *BAX* in EqASC_EMS_ cultures, while increasing the mRNA expression of anti-apoptotic *BCL-2*. Further, the *BAX* transcript level increased following metformin treatment in EqASC_HE_ cultures, however, the *BAX*/*BCL-2* ratio decreased, indicating an anti-apoptotic effect correlating with low ROS activity in those cultures (Figure 4e–g).

### 3.4. Metformin Induces Intracellular Accumulation of β-Catenin and Wnt-3a, However, Only in EqASCs Derived from EMS Horses

The expression of WNT-3a and β-catenin was analysed on mRNA and protein levels. The mRNA level for β-catenin increased following metformin treatment both in EqASC_HE_ and EqASC_EMS_. We also noted that metformin increases the transcript level of *WNT-3A*, however, only significantly in EqASC_EMS_ cultures. The intracellular accumulation of Wnt-3a and β-catenin was also significantly higher in EqASC_EMS_ (Figure 5).

### 3.5. Metformin Decreases miR-16-5p, miR-21-5p, miR-29a-3p, miR-140-3p and miR-145-5p Levels in EqASCs Derived from EMS Horses

miRNA analysis showed that metformin does not influence *miR-16-5p*, *miR-21-5p* and *miR-29a-3p* levels in EqASC_HE_, while the levels of *miR-140-3p* and *miR-145-5p* in those cultures were increased. All tested miRNAs were significantly downregulated in EqASC_EMS_ cultures (Figure 6).

## 4. Discussion

Adipose tissue is defined by heterogenous cellular composition that may be altered during disease conditions, such as diabetes and equine metabolic syndrome [46,47,48,49]. The morphology of mature adipocytes is also influenced by physiological conditions and might change under specific medication treatment, e.g., metformin was shown to reduce the diameter of adipocytes as well as to induce a greater heterogeneity of the tissue [50]. Currently, much attention is paid to the cellular composition of stromal vascular fraction (SVF) obtained from adipose tissue. This fraction contains a plethora of cells, including endothelial cells, fibroblasts, B- and T-lymphocytes, macrophages, myeloid cells, pericytes, pre-adipocytes, smooth muscle cells and, finally, culture-adherent adipose stromal cells (ASCs) [51]. Due to high cellular plasticity and enhanced self-renewal, ASCs are considered an excellent therapeutic tool in cell-based therapies for various disorders, including diabetes and the metabolic syndrome [46]. Moreover, the great pro-regenerative potential of ASCs also relies on their paracrine activity and immunomodulatory properties [52,53].

We have previously shown that EMS affects various aspects of ASC cellular activity and limits their clinical application. Generally, ASCs derived from horses with EMS (EqASC_EMS_) exhibit lower proliferative and metabolic potential. Moreover, cultures of EqASC_EMS_ are characterised by increased senescence and cell death. Further, we noticed imbalance of the oxidative status in that EqASC_EMS_ was related to endoplasmic reticulum (ER) stress and deterioration of mitochondrial dynamics [2,3,4,5,41]. Currently, various pretreatment conditions and culture strategies are applied in terms of improvement of the regenerative potential of multipotent stromal cells residing in different tissue niches [52,54]. We previously found that combination of 5-azacytydine and resveratrol (AZA/RES) has a favourable influence on the proliferation, viability and multipotency of EqASC_EMS_ [20]. The mechanism of 5-azacitidine (5-AZA) is related to inhibition of DNA methyltransferase (DNMT), while resveratrol was recognised, inter alia, as an activator of AMPK/PGC-1α signalling, improving mitochondrial biogenesis and dynamics [20]. The anti-aging and senolytic action of AZA/RES combination has been demonstrated towards progenitor cells of adipose origin. In the experiment, we focused on another AMPK activator, i.e., metformin, and its potential effect on the basic cytophysiological features of ASCs, including proliferation, metabolism and viability. The metformin was described as an agent with pleiotropic activity, which also includes anti-aging and senolytic activity [55]. Recently, we have observed growing interest in metformin and its application as a pro-regenerative molecule [56].

Our current research shows that metformin exerts pro-proliferative effect towards EqASCs derived from both healthy and EMS-affected horses (EqASC_HE_ and EqASC_EMS_, respectively)_._ We noted increased proliferation and metabolic activity after metformin treatment, as well as shift of EqASCs to the S-phase of the cell cycle. The results are consistent with our previous studies applying an ex vivo model and showing the pro-proliferative effect of metformin on mouse ASC (mASCs) and progenitor cells isolated from olfactory bulb [21,22]. Previously, recognising the pro-regenerative potential of metformin, we used it as a bioactive molecule for functionalisation of sol–gel coatings covering metallic implants. We tested cytocompatibility of the obtained biomaterials using a model of human ASCs (hASCs) [57]. In this experiment, we found that metformin may enhance proliferation of cells, shorten the population doubling time, and improve metabolic activity. Earlier, the pro-proliferative effect of metformin was also established in terms of MSCs derived from bone marrow (BMSC) and osteoblast progenitors [58,59], as well as adipose-derived stromal cells [60]. Metformin generally acts in dose- and time-dependent manner, therefore, consideration of proper metformin dosage is crucial in terms of obtaining a desirable effect [50,61]. Metformin is also known as an anticancer drug. We previously tested metformin at concentrations that inhibited proliferation of various cancer cell lines, including breast, ovarian and pancreatic [62]. Our results showed that metformin in higher concentrations exerts an anti-proliferative effect towards mASCs [50] and mBMSCs [61].

We also indicated an improved oxidative status of mASCs derived from animals treated with metformin. This was correlated with reduction of reactive oxygen species (ROS) and nitric oxide (NO), and increase of SOD (superoxide dismutase) activity [21]. In the current study of EqASCs, we have confirmed the antioxidant effect of metformin. We also show that metformin improves mitochondrial membrane activity. The ultrastructural observations of metformin indicated enhanced dynamics of mitochondria in cultures derived from EMS horses. In EqASC_EMS_, we have observed elongated mitochondria. Mitochondria elongation occurs during macroautophagy, which is a mechanism allowing to sustain cellular ATP production and viability of cells [63]. These results correlate with increased viability of EqASCs in cultures treated with metformin, and are in agreement with studies performed by Wang et al., who showed that metformin can protect H9c2 cells against hyperglycaemia-induced apoptosis and Cx43 downregulation through the induction of the autophagy pathway [64].

Analysis of the apoptosis profile revealed that metformin may act as anti-apoptotic agent towards EqASCs derived both from healthy and EMS horses. The results correlate with our previous findings, showing increased viability of mouse progenitor cells derived from animals treated with metformin [21,22]. However, the results are in contradiction to data presented by He et al. [65], who showed that metformin significantly induces apoptosis of MSCs isolated from human umbilical cord, even at 0.1 mM concentration. He et al. showed that metformin induces apoptosis in cells in dose- and time-dependent manner, and indicated that metformin at 2 mM concentration induced a sub-G1 peak, which is a suggestive marker of apoptosis [65]. We confirmed the anti-apoptotic activity of metformin in EqASC cultures at different levels. Firstly, by the distribution of cells in the cell cycle, showing a reduced percentage of cells in G0/G1-phase; secondly, by depicting increased mitochondrial membrane potential following metformin treatment; and, finally, determining the apoptosis profile using annexin V/PI and calcein/PI staining. Our results clearly indicate that metformin improves EqASC viability. Additionally, we noted increased transcript levels of the anti-apoptotic *BCL-2* gene, in contrast to the findings of He et al. [65]. The discrepancy between our findings and those of He et al. may be due to the model used, as well as different metformin concentrations. He et al. performed a comprehensive analysis of the pro-apoptotic effect of metformin using dosages above 1 mM, which is in agreement with our previous studies [50,61].

Due to the fact that Wnt/β-catenin signalling was indicated as a crucial pathway in terms of modulating MSC self-renewal and differentiation, we were interested in its expression profile after metformin treatment. It has been shown that fine-tuning of Wnt/β-catenin signalling coordinates MSC function [66]. Particularly, Kim et al. (2015) revealed that low levels of β-catenin in MSCs result in increased expression of genes involved in cell cycle control and DNA metabolism, while high levels of β-catenin were linked to increased expression of genes crucial for development and metabolism [66]. Additionally, Subramaniam et al. [32] tested the influence of AICAR and metformin on hepatic stellate cells (HSCs). The study showed that both agents activate AMPK signalling in quiescent HSCs, but elicit distinct effects on cells function. Interestingly, AICAR rapidly induced cell death of HSCs, while the cells remained viable after metformin treatment. Metformin induced activin membrane-bound inhibitor (Bambi) and activated a pro-survival Wnt/β-catenin signalling pathway [32]. Wnt pathway can also be regulated by ROS levels, which indicates crosstalk between Wnt and redox signalling. It has been shown that low levels of ROS activate Wnt signalling and improve differentiation of MSCs towards osteogenic cells [67]. Our results confirm that lowered levels of ROS may induce intracellular accumulation of WNT. Increased expression of WNT in cells treated with metformin may also explain the pro-osteogenic action of the drug towards ASCs, which was emphasised in our previous studies [21,57].

Metformin was also found to modulate microRNA levels. The miRNA profile following metformin treatment has previously mainly been established for cancer cell lines [68,69]. Here, we measured the levels of the following miRNAs: *miR-16-5p*, *miR-21-5p*, *miR-29a-3p*, *miR-140-3p* and *miR-145-5p*, that previously have been reported relevant for self-renewal and differentiation of mesenchymal stem cells [70,71]. We found that, in EqASC_EMS_ cultures, expression of *miR-16-5p*, *miR-21-5p* and *miR-29a-3p* decreased following metformin treatment. miR-16-1 level has been reported inversely correlated to BCL-2 expression in chronic lymphocytic leukaemia (CLL) [72]. Indeed, our data indicate increased expression of *BCL-2* in cultures treated with metformin. Further, it has been reported that miR-16 controls myoblast proliferation and apoptosis via coordinated regulation of BCL-2 activation [72]. It has also been shown that overexpression of mir-21 is related to increased proliferation activity of BMSCs [73]. This also correlated to increased expression of BCL-2 and vascular endothelial growth factor (VEGF), and decreased expression of BAX. This is different to the profiles observed in our model. Firstly, we observe constitutive expression of miR-21-5p in EqASCs derived from healthy horses. Secondly, native cultures of EqASC_EMS_ are characterised by increased occurrence of apoptosis and increased expression of *BAX* transcripts, which is associated to accumulation of *mir-21-5p*. Upon metformin treatment, the pattern was reversed: we found reduced levels of *mir-21-5p* and *BAX*, while *BCL-2* level was increased. It seems that regulation of mir-21-5p is complex, and it may influence the differentiation process of BMSCs [74]. Contradictory data exist concerning mir-21 function as a regulator of MSC fate and lineage commitment. For example, it has been shown that overexpression of mir-21 is related to osteoclastogenesis [75], adipogenesis [76], and osteolysis [77]. Nevertheless, it was also reported that rat BMSCs overexpressing mir-21 accelerate fracture healing in a rat closed femur fracture model [74].

Further, miR-29a-3p, in EqASCs, was found to have the same expression pattern as what we observed for miR-21-5p. Overexpression of miR-29a was correlated to reduced levels of Slit glycoprotein 2 (SLIT2) and its receptor Roundabout 1 (ROBO1) which, in turn, resulted in inhibition of mesenchymal stem cell viability and proliferation [78]. Moreover, it was shown that transfection of miR-29 family members at an early stage of somatic cell reprogramming may decrease the number of colonies expressing pluripotent markers, such as *Oct4* [79]. These results may explain the mir-29a-3p expression profile observed in EqASC_EMS_ treated with metformin. These cultures were characterised by higher proliferative activity and viability when compared to non-treated cells characterised by increased levels of *mir-29a-30*.

miR-140-3p and miR-145-5p in EqASC_HE_ and EqASC_EMS_ were differentially modulated by metformin. Following treatment, we observed increased levels of both transcripts in EqASC_HE_, while miR-140-3p and miR-145-5 levels in EqASC_EMS_ cultures were lower. mir-140 has been reported to modulate proliferation and differentiation of human dental pulp stem cells (DPSCs) [80]. Overexpression of miR-140-5p improved proliferation of DPSCs and aggravated their differentiation, whereas suppression of miR-140-5p had the opposite effect. Similarly, higher level of mir-145-5p was related to the lower ability of MSCs to undergo chondrogenic differentiation. mir-145-5p is generally considered suppressor of cell growth, and its reduced level in EqASC_EMS_ cultures may explain their increased proliferative activity. Moreover, we have previously shown that microvesicles isolated from EMS contained high levels of mir-140 [5]. mir-140 has been reported marker for type II diabetes (T2D) [81]. Individuals with T2D have mir-140-3p levels upregulated compared to individuals with gestational diabetes mellitus, and downregulated compared to individuals with type I diabetes mellitus [82].

Today, miRNA is in focus, and has been proposed to serve as new biomarker for the diagnosis and treatment of metabolic syndrome in horses [81]. Identification of useful miRNA signatures in horses is an emerging field [83], and highly specific and sensitive methods, such as the two-tailed RT-qPCR used here, are vital to establish their usefulness as biomarkers for various horse diseases, including EMS.

## 5. Conclusions

Our results indicate that metformin improves proliferative activity of EqASCs derived from healthy and EMS horses. Metformin enhances mitochondrial metabolism reducing the percentage of cells with low mitochondrial membrane potential, which was related to the increased viability of EqASCs. Following metformin treatment, accumulation of WNT-3A and β-catenin was observed in EqASC_EMS_. Metformin also modulates the expression of several miRNAs associated with cell proliferation, viability and differentiation. Bearing in mind that metformin may also promote differentiation of MSCs, it may be reasonable to test this drug as a preconditioning agent in osteogenic, chondrogenic, as well as adipogenic cultures of EqASCs.

## Figures and Tables

**Figure 1 cells-08-00080-f001:**
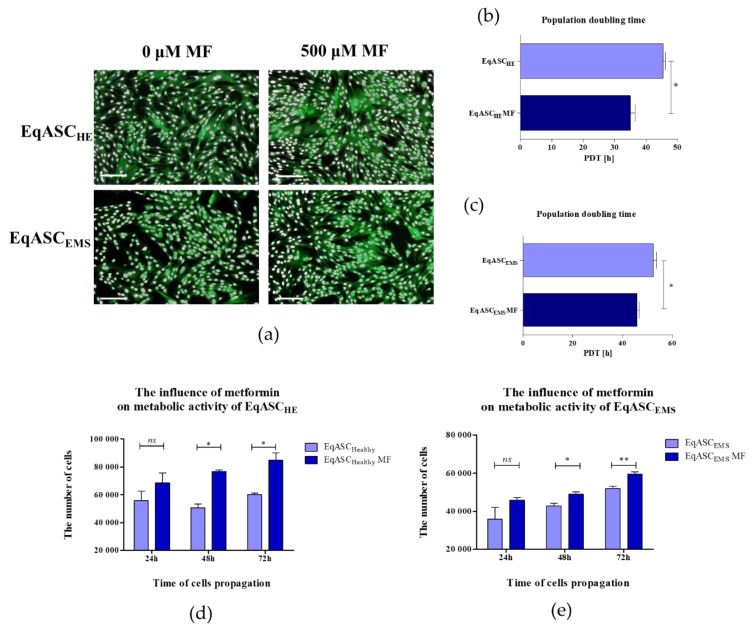
The influence of metformin on proliferative and metabolic activity of adipose-derived stromal cells isolated from horses (EqASCs). The proliferative activity was evaluated based on microphotographs obtained with epifluorescence microscope—scale bar 250 μm (**a**) population doubling time ratio (**b**,**c**) and metabolic activity (**d**,**e**). The statistically significant changes were indicated with asterisks; * *p* < 0.05; ** *p* < 0.01, while non-significant differences are marked as *ns*.

**Figure 2 cells-08-00080-f002:**
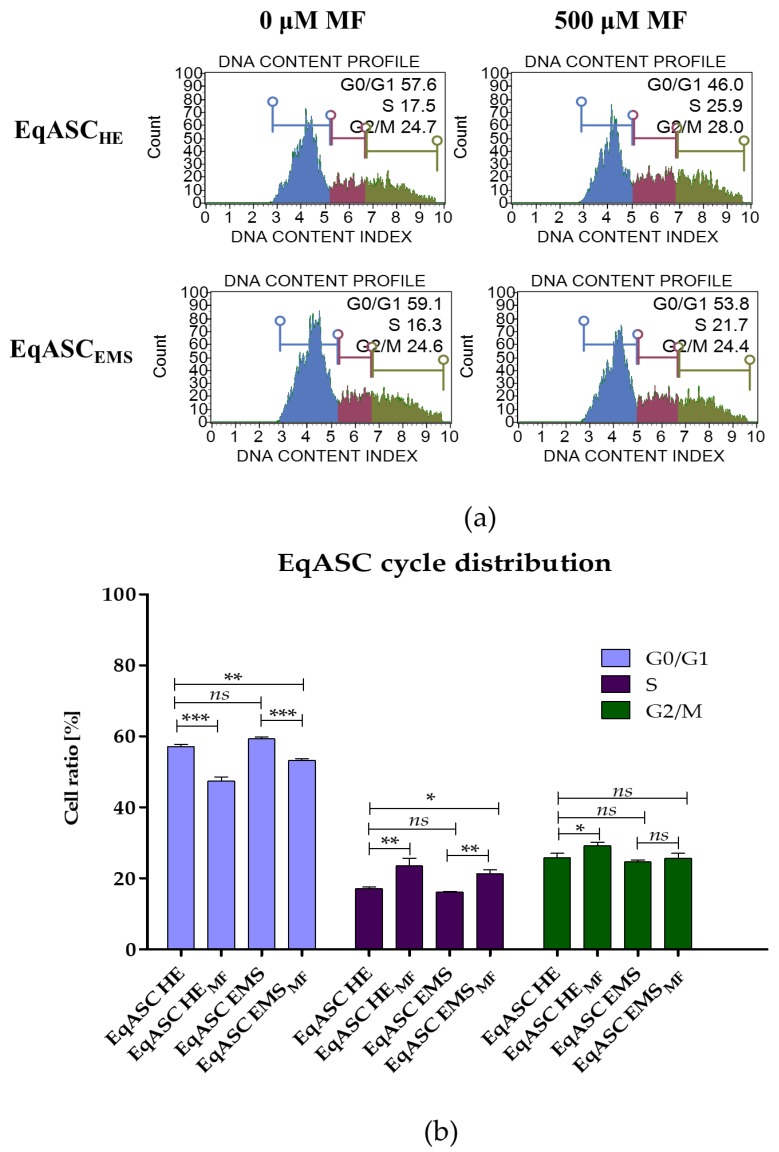
The influence of metformin on the distribution of EqASCs during the cell cycle, with representative histograms (**a**) and results of statistical analysis (**b**). The statistically significant changes are indicated with asterisks; * *p* < 0.05; ** *p* < 0.01 and *** *p* < 0.001. Non-significant differences are marked as *ns*.

**Figure 3 cells-08-00080-f003:**
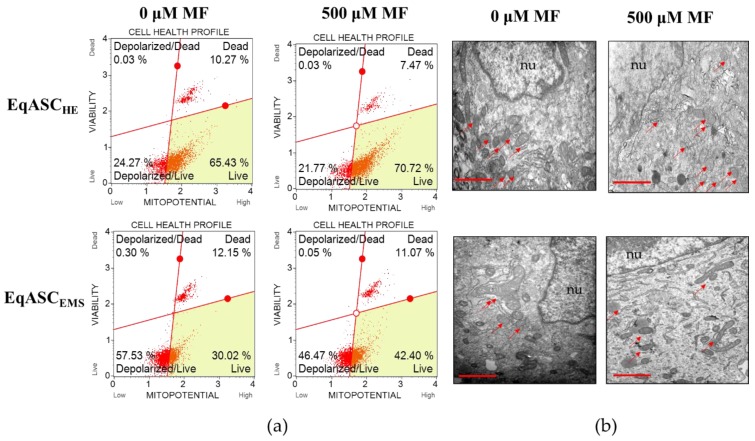
The influence of metformin on mitochondrial membrane potential and oxidative. Representative dot plots indicating distribution of cells accordingly to the mitochondrial membrane potential (**a**). The ultrastructure of EqASC_HE_ and EqASC_EMS_ without metformin treatment and after metformin treatment. Mitochondria are indicated with red arrows, and nuclei with *nu* symbol, scale bar: 2 μm (**b**). Analysis of cell viability based on mitochondrial potential (**c**). Comparison of total depolarised cells in tested experiments (**d**). Evaluation of reactive oxygen species (**e**), nitric oxide (**f**) and superoxide dismutase activity (**g**). Significant changes are indicated with asterisks: * *p* < 0.05; ** *p* < 0.01 and *** *p* < 0.001, while non-significant differences are marked as *ns*.

**Figure 4 cells-08-00080-f004:**
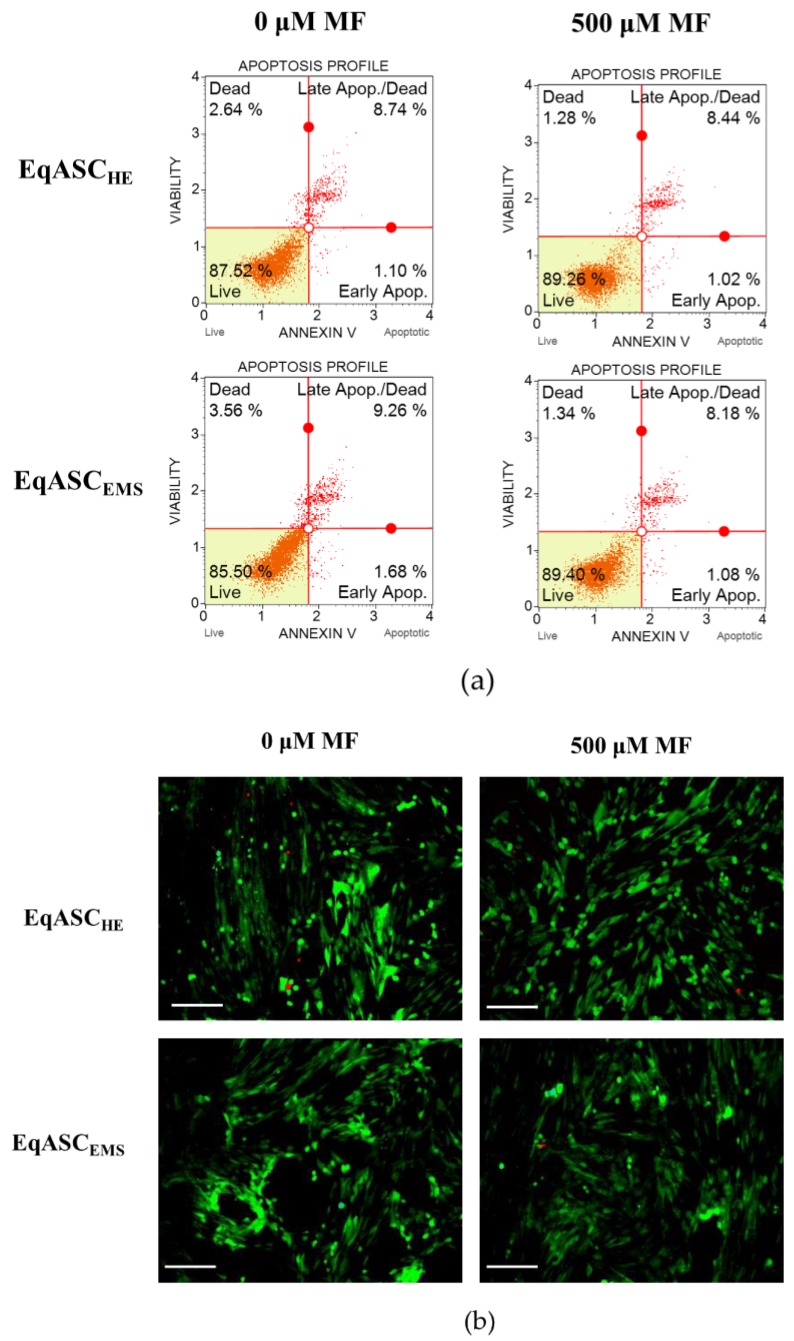
The influence of metformin on the apoptosis profile. Representative dot plots indicating distribution of cells after annexin V/ propidium iodide staining (**a**) Representative images obtained after calcein/PI staining, scale bar—250 μm. (**b**) Analysis of cell viability (**c**) and apoptosis (**d**). Measured transcript levels for *BAX* (**e**) *BCL-2* (**f**) and their ratio (**g**) Statistically significant changes are indicated with asterisks: * *p* < 0.05; ** *p* < 0.01 and *** *p* < 0.001, while non-significant differences are marked as *ns*.

**Figure 5 cells-08-00080-f005:**
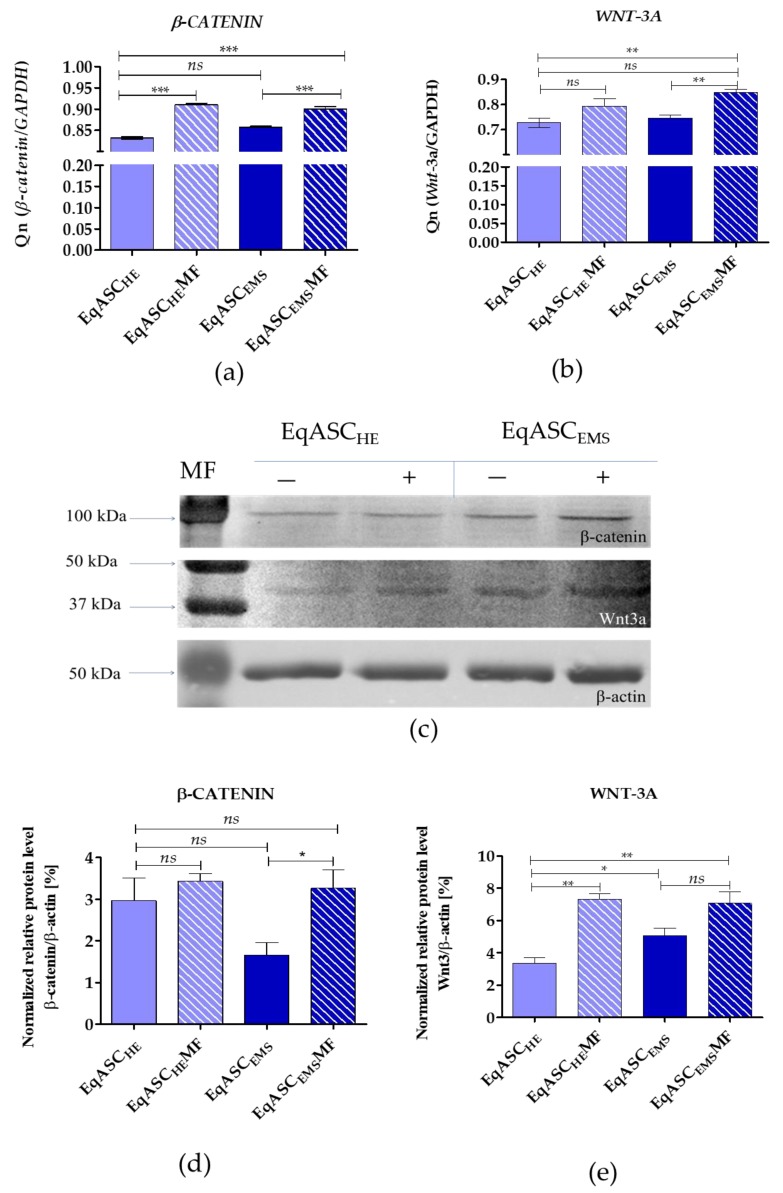
Determination of Wnt-3/β-catenin expression (**a**,**b**) and corresponding intracellular accumulation of protein (**c**–**e**). Statistically significant changes are indicated with asterisks: * *p* < 0.05; ** *p* < 0.01 and *** *p* < 0.001, while non-significant differences are marked as *ns*.

**Figure 6 cells-08-00080-f006:**
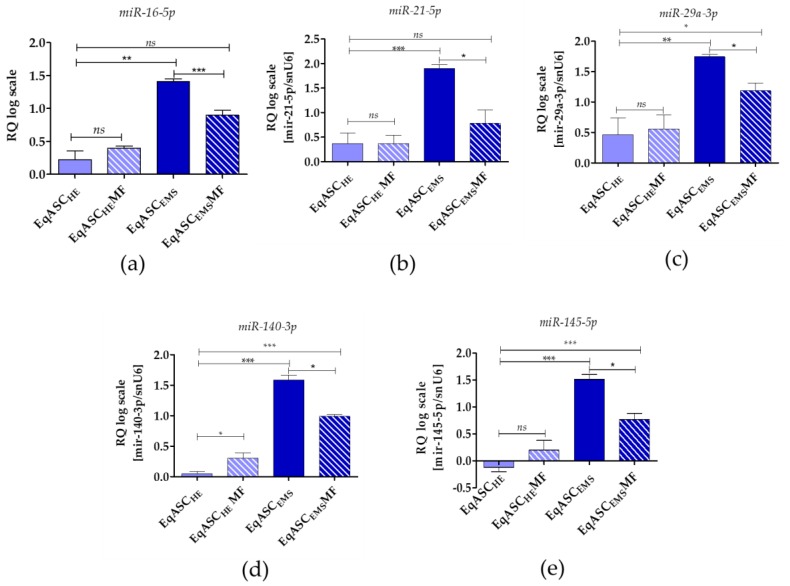
miRNA expression measured with two-tailed RT-qPCR. The following miRNAs were tested: miR-16-5p (**a**), miR-21-5p (**b**), miR-29a-3p(**c**), miR-140-3p (**d**) and miR-145-5p (**e**). Data are normalised to U6 snRNA levels and expressed as fold changes compared to the sample with the lowest level. Statistically significant changes are indicated with asterisks: * *p* < 0.05; ** *p* < 0.01 and *** *p* < 0.001, while non-significant differences are marked as *ns*.

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
