# Peer review of "Metformin Increases Proliferative Activity and Viability of Multipotent Stromal Stem Cells Isolated from Adipose Tissue Derived from Horses with Equine Metabolic Syndrome"

_cells, 2019, doi:10.3390/cells8020080_

Round 1

Reviewer 1 Report

In this manuscript “Metformin increases proliferative activity and viability of multipotent stromal stem cells isolated from adipose tissue derived from horses with equine metabolic syndrome” by Smieszek et al, the authors analyzed the role of MET on proliferation and viability of adipose-derived stromal cells isolated from healthy horses (EqASC) and horse affected by equine metabolic syndrome (EqASCEMS). The authors also discussed about effects of MET on WNT3a single transduction pathway. Finally, authors established that MET as a compound, which has capacity to improve the cellular health of EqASCEMS. This is a well conducted study and the data shown support the conclusion reached. The concept and outcome of manuscript is very interesting and I do not have any comments related to this manuscript.

Author Response

Wroclaw, 31st of December 2018

Dear Reviewers,

We would like to thank you for all your comments and suggestions. We are grateful for careful evaluation of this paper and accurate summary of our paper. We are pleased to address all your concerns for revision. We hope that the answers will meet with your approval, and we trust that this revised version of our manuscript is clear and concise as well as scientifically strengthened. We think that thanks to your accurate recommendations our manuscript gained new quality. All recommendations will be taken into account when writing further articles.

The revisions were highlighted using the "Track Changes" function in Microsoft Word.

We hope you will be satisfied with our manuscript revision and eventually find the article suitable for publication.

Yours sincerely,

Agnieszka Śmieszek

On a behalf of the Authors

Dear Reviewer 1,

We are very pleased that our article interested you. Thank you for your kind summary. We are really glad that you accepted the paper as suitable for publication. The language  of the manuscript was improved.

Reviewer 2 Report

In the current study, the authors explored that the Metformin increases proliferative activity and viability of multipotent stromal stem cells isolated from adipose tissue derived from horses with equine metabolic syndrome. However, I have following concerns related to this study:

a. Authors have already demonstrated enhanced proliferative and osteogenic effects of metformin on ADSC isolated from mice (Oxid Med Cell Longev. 2016; 2016: 9785890, DOI:  10.1155/2016/9785890).

Authors have also noted anti-oxidant and anti-apoptotic activities of metformin through decreased ROS levels. (Int. J. Mol. Sci. 2017, 18(4), 872; doi:10.3390/ijms18040872).

Besides, other below mentioned studies have also revealed positive effects on stem cell characteristics:

1. Metformin increase proliferation of stem cells cultured on biphasic bone granules (https://www.sciencedirect.com/science/article/pii/S0003996918303923; DOI: https://doi.org/10.1016/j.archoralbio.2018.07.012)

2. Effect of metformin on dental pulp stem cells attachment, proliferation and differentiation cultured on biphasic bone substitutes. (Arch Oral Biol. 2018 Nov;95:44-50. doi: 10.1016/j.archoralbio.2018.07.012. Epub 2018 Jul 18)

Authors have concluded that metformin may promote MSC differentiation, and therefore could tested as a preconditioning agent in osteogenic, chondrogenic as well as adipogenic cultures of EqASCs. These effects have already been demonstrated by authors in their previous studies (J. Clin. Med. 2018, 7(12), 482; doi:10.3390/jcm7120482) (Oxid Med Cell Longev. 2016;2016:9785890. doi: 10.1155/2016/9785890. Epub 2016 Apr 18).

These evidences are clearly an indicative of positive effect of metformin, therefore, this research work lack sufficient novelty.

b. Figure 5c: The band size of actin (Sigma Aldrich, A2066) should be close to 42kD, while author have indicated at 50kD.

c. Most of figures are of not high quality, and titles of most of graphs are blurred/unclear.

d. Figure 1d, colors boxes indicating study groups seems to be cut.

d. In figure legends 5 and 6, (**) cannot indicate both p<0.01 and p<0.001.

e. Authors have cited reference no. 9 which has been downloaded from researchgate. It could have been cited directly.

f. English writing in many sentences is incorrect, such as

Line 503-505

“Metformin enhances mitochondrial metabolism reducing the percentage of cells with low mitochondrial membrane potential, which increases viability of EqASCs, and It accumulates WNT-3A and β-catentin in EqASCEMS.”

 Line 363-365

“Microscopic evaluation of EqASCHE and EqASCEMS cultures showed that metformin does not affectd the cells’ morphology - the cells maintain proper fibroblast-like morphotype.”

In addition to this, the manuscript has many grammatical and spelling errors.

Author Response

Wroclaw, 31st of December 2018

Dear Reviewers,

We would like to thank you for all your comments and suggestions. We are grateful for careful evaluation of this paper and accurate summary of our paper. We are pleased to address all your concerns for revision. We hope that the answers will meet with your approval, and we trust that this revised version of our manuscript is clear and concise as well as scientifically strengthened. We think that thanks to your accurate recommendations our manuscript gained new quality. All recommendations will be taken into account when writing further articles.

The revisions were highlighted using the "Track Changes" function in Microsoft Word.

We hope you will be satisfied with our manuscript revision and eventually find the article suitable for publication.

Yours sincerely,

Agnieszka Śmieszek

On a behalf of the Authors

Review 2

Answer “a.”

Thank you for this comment. Indeed our group is interested in metformin influence on cytophysiology of progenitor cells – their proliferation and differentiation, yet these results show the effect of metformin on equine adipose-derived stromal cells (EqASC). Previously we analysed the role of metformin using smaller model organisms (i.e. mice and rat) and therefore we believe our research is not devoid of novelty. Our research is the first one to show the effect of metformin on EqASC derived from horses with equine metabolic syndrome. The metabolism, proliferation and differentiation of these cells is impaired, thus the effect of metformin could differ from that observed using model of mice and rat ASC. The first concern in terms of EqASCEMS culture is their lowered proliferative potential and viability affecting the regenerative/therapeutic potential of autologous transplants which are currently widely applied for the treatment of muscoskeletal diseases in horses. Bearing in mind our previous study which showed pro-proliferative action of metformin, our natural curiosity was to test whether metformin influences proliferation and metabolism of EqASCEMS. We believe our study will be interesting for researchers that are focused on biology of adipose-derived stromal cells, especially of equine origin. At the same, we agree that there are many proofs (including our own research) of positive effect of metformin on MSCs’ proliferative status. However due to some differences in terms of models and metformin concentrations used, some discrepancies in cell response are also noted [1–3].

[1]          Sedlinsky C, Molinuevo MS, Cortizo AM, et al. Metformin prevents anti-osteogenic in vivo and ex vivo effects of rosiglitazone in rats. European Journal of Pharmacology 2011; 668: 477–485.

[2]          Śmieszek A, Basińska K, Chrząstek K, et al. In Vitro and In Vivo Effects of Metformin on Osteopontin Expression in Mice Adipose-Derived Multipotent Stromal Cells and Adipose Tissue. Journal of Diabetes Research. Epub ahead of print 2015. DOI: 10.1155/2015/814896.

[3]          Montazersaheb S, Kabiri F, Saliani N, et al. Prolonged incubation with Metformin decreased angiogenic potential in human bone marrow mesenchymal stem cells. Biomed Pharmacother 2018; 108: 1328–1337.

Furthermore, we believe the novel element in the data presented is also related to the method of miRNA detection. The two-tailed method was used for the first time to determine the levels of miRNA in EqASC.

Answer “b”

Thank you for pointing this out. Yes, we confirm that the actin detected in in cell lysates derived from horses using A2066 antibody (Sigma Aldrich) has molecular weight of ~50 kDa. This is true and repetitive result. Aberrant migration on SDS-PAGE is a well-known phenomenon that can result from the number of SDS-molecules that can attach to a particular protein. The actin is a protein cells have in abundance, so the differences in the migration can also result of overloading the gel. It can be also the result of post-translational modifications that we are not aware of. Protein can undergo glycosylation and lipid modifications, all of which can have a strong influence on a proteins migration on SDS-PAGE.

Unfortunately, vast majority of publications indicated on Sigma Aldrich site as a reference to A2066 product(https://www.sigmaaldrich.com/catalog/product/sigma/a2066?lang=pl&region=PL&gclid=EAIaIQobChMI2dCBgvfC3wIVGouyCh3kSwnUEAAYASAAEgKmJPD_BwE ) show only actin band, not identified in terms of its MW of actin (accordingly to molecular mass marker). Such presentation of data makes it difficult to compare. Nevertheless we found publication by Wu et al (Nature Communications volume 7, Article number: 10533 (2016)) who also used the A2066 antibody for detection of actin in HEK cell line, and identified bands ~ 50 kDa (Figure 1 and 2).

Answer “c”

Please accept our apologies for that. All Figures were provided as separate (editable) files during submission process. We were hoping that you had access to them. Indeed, we see that in the manuscript all Figures are smaller and of poor quality. Due to the space limit we indicated the location of the figures within the text and believed that you will be able to compare them to the original Figures. In the current version, we improved the size of the figures. We hope that it will find your acceptance. A

Probably during the process of manuscript preparation the editable files will be used, then the figures will be in high resolution.

Answer “d”

Thank you for noticing this. The Figure 1d was replaced. Corrected Figure 1 will be also provided in an editable form.

Answer “d-prim”

Thank you for flagging that. Of course this is a mistake. It was corrected. The proper form is (**)p<0.01 and (***)p<0.001.

Answer “e”

Yes. The reference was corrected.

Answer “f”

We have read manuscript carefully and corrected when necessary.

Reviewer 3 Report

Dear colleagues!

As a Reviewer assigned by the Editorial board to assess the manuscript "Metformin increases proliferative activity and viability of multipotent stromal stem cells isolated from adipose tissue derived from horses with equine metabolic syndrome" by Smieszek et al. I have the following comments:

1) certain points were given as comments in the PDF attached with suggestion of improvements

2) overall, the main point that leaves the Reader with insufficient information is lack of physiological outcome in the study. E.g., assessment of adipogenesis of ASC from EMS-affected and healthy animals and influence of tested substance (Metformin) on this process. Furthermore, efficacy of adipogenesis itself may be restored by Metformin yet in the Introduction the authors cite literature describing changes in ASC during inset of EMS (insulin-resistance, inflammatory response etc), so to fully support the concept it would be valuable to:

a) assess eqASC adipogenic potential (and influence of Metformin as well)

b) describe and assess signs of EMS and EMS-related changes in adipocytes derived from ASC (and influence of Metformin as well).

Overall, scientific merit of the paper is potentially high and I would suggest improvements to strengthen scientific level of the work. 

Author Response

Wroclaw, 31st of December 2018

Dear Reviewers,

We would like to thank you for all your comments and suggestions. We are grateful for careful evaluation of this paper and accurate summary of our paper. We are pleased to address all your concerns for revision. We hope that the answers will meet with your approval, and we trust that this revised version of our manuscript is clear and concise as well as scientifically strengthened. We think that thanks to your accurate recommendations our manuscript gained new quality. All recommendations will be taken into account when writing further articles.

The revisions were highlighted using the "Track Changes" function in Microsoft Word.

We hope you will be satisfied with our manuscript revision and eventually find the article suitable for publication.

Yours sincerely,

Agnieszka Śmieszek

On a behalf of the Authors

Dear colleagues!

As a Reviewer assigned by the Editorial board to assess the manuscript "Metformin increases proliferative activity and viability of multipotent stromal stem cells isolated from adipose tissue derived from horses with equine metabolic syndrome" by Smieszek et al. I have the following comments:

1)     certain points were given as comments in the PDF attached with suggestion of improvements:

a)     MET - Abbreviation is typically used for Mesenchymal-epithelial transition

Answer 1a: Thank you. We corrected this and used proper abbreviation for metformin i.e. MF.

b)    “as an urgent issue”  - Typically term urgent means "life-threatening status requiring immediate medical intervention", e.g. peritoneal syndromes or system acute infection while metabolic syndromes in most mammals are typically long-lasting chronic states requiring urgent aid only in case of decompensation with severe hyperglycemia

Answer 1 b: Yes, we agree that the word “urgent” was not adequate in this sentence. We change it on “burning issue”.

c)     G2M graph - Difference marked by ** does not look significant - bards and error "whiskers" are almost identical. Please, assure or clarify this point

Answer 1 c:

Yes, thank you for pointing this out. We are sorry that we did not notice that. This mistake was unintended. The file included in the manuscript must have been saved without proper changes. Indeed, we do not observed the statistical difference between group EqASCHE and EqASCEMS as well as EqASCHE and EqASCEMS treated with MF. This was clarified in the current version.

d)    Major points

1)     Please clarify whether it was possible to assess and "induced" apoptosis besides the "basal" level described in the section. Induction of apoptosis in MSC or any other modeling of "unfavorable conditions" will improve the scientific strength and show protective effects of Metformin.

2)     BAX/BCL2 assessed by qPCR might be strengthened by Western-blotting (e.g. as in section 3.4 for wnt and b-catenin.

Answer 1d:

(1)  Thank you for this suggestion. We agree that such model would improve the impact of the presented data. Certainly we would like to perform such study in the future. We would like to induce the oxidative stress and apoptosis in ASC and determine the effects of metformin in terms of recovery of cells functions.  In this study “unfavourable conditions” were related with the EMS condition that significantly affects the cytophysiology of progenitor cells, what was previously described by us [1].

[1]       Marycz K, Kornicka K, Basinska K, et al. Equine Metabolic Syndrome Affects Viability, Senescence, and Stress Factors of Equine Adipose-Derived Mesenchymal Stromal Stem Cells: New Insight into EqASCs Isolated from EMS Horses in the Context of Their Aging. Oxidative Medicine and Cellular Longevity. Epub ahead of print 2016. DOI: 10.1155/2016/4710326.

(2)  Yes, that is true. Currently, we do not have the antibodies for equine BAX/BCL-2, but we plan to purchase it. We are sorry, but won’t be able to improve the result within 10 days given for the revision of our paper.

e)     typo error, space missing

Answer 1e:

Thank you. Corrected.

f)      As a suggestion "off-review" you might be interested in the following transcriptional mechanism (involving wnt and glucose-insulin regulation):

https://www.ncbi.nlm.nih.gov/pubmed/27815072

https://www.frontiersin.org/articles/10.3389/fendo.2018.00346/full

Some of them are liver-specific yet ASC may present a good object for further investigation.

Answer 1f:

Thank you for this valuable comment and links for those interesting publications.

Yes, we agree that indicated transcriptional mechanisms are good object for investigation. Currently, we are focused on determining mutations in peroxisome proliferative-activated receptor γ (PPARγ) in EMS horses. Indeed, accordingly to the recent discoveries the Prep1 may be good target in terms of improving and/or treating metabolic diseases. Once again thank you for sharing this idea with us.

2) overall, the main point that leaves the Reader with insufficient information is lack of physiological outcome in the study. E.g., assessment of adipogenesis of ASC from EMS-affected and healthy animals and influence of tested substance (Metformin) on this process. Furthermore, efficacy of adipogenesis itself may be restored by Metformin yet in the Introduction the authors cite literature describing changes in ASC during inset of EMS (insulin-resistance, inflammatory response etc), so to fully support the concept it would be valuable to:

a) assess eqASC adipogenic potential (and influence of Metformin as well)

b) describe and assess signs of EMS and EMS-related changes in adipocytes derived from ASC (and influence of Metformin as well).

Overall, scientific merit of the paper is potentially high and I would suggest improvements to strengthen scientific level of the work. 

Answer 2:

This is a really accurate comment and suggestion. In the current study, we decided to determine the influence of metformin on proliferation and viability of ASC derived from EMS-affected and healthy animals.  We plan to continue this subject and study the influence of metformin on progenitor cells from EMS horses. In further studies we will analyse the influence of metformin on adipogenesis, but also plan to verify metformin role in the process of osteo- and chondrogenesis. Taking into account the Reviewers’ suggestion, we are now sure that this is a good lead and a promising topic that may interest many researchers.  Thank you for appreciating the scientific merit of the paper. 

Round 2

Reviewer 1 Report

no comments.

Author Response

Dear Reviewer,

Thank you for accepting the paper in a current form.

Yours sincerely,

Agnieszka Śmieszek

Reviewer 3 Report

Dear colleagues!

As a reviewer assigned to the manuscript I fully accept your comments and rebuttal made for majority of the points yet for point 2

2) overall, the main point that leaves the Reader with insufficient information is lack of physiological outcome in the study. E.g., assessment of adipogenesis of ASC from EMS-affected and healthy animals and influence of tested substance (Metformin) on this process. Furthermore, efficacy of adipogenesis itself may be restored by Metformin yet in the Introduction the authors cite literature describing changes in ASC during inset of EMS (insulin-resistance, inflammatory response etc)

I find it of crucial importance to show that outcome can be altered by MF in EMS-affected ASC.

I leave the point for Editorial decision as far as you have highlighted 10 days fore revision, thus, you can hardly complete a significant set of experiments (especially adipogenic differentiation which takes 15-20 days) under these terms.

Regards, Reviewer

Author Response

Dear Reviewer,

The Editor suggested to answer to your comment once again, in more appropriate manner.

Yes, we emphasized in the introduction section that cytophysiology of ASC derived from EMS is altered. We have previously showed that ASCs from EMS subjects are characterized by lower proliferative activity, associated with reduced clonogenic potential as well as lowered expression of Ki-67. Moreover, the metabolic activity of ASCs is also attenuated by EMS and involves deterioration of mitochondrial dynamics, which is related to lowered mitochondrial metabolism and induced macroautophagy process. Due to the fact that therapeutic application of MSC is influenced by proliferative and metabolic status, the first set of experiments was aimed at determination of metformin effect on metabolic and proliferative activity of EqASCEMS. In this paper we showed that metformin improves the proliferation of ASC from EMS-affected horses shortening their population doubling time and promoting S phase entry. Further, following metformin treatment we observed the increased metabolic activity of EqASCEMS, which was accompanied by increased mitochondrial membrane potential and cells viability, reduced oxidative stress as well as intracellular accumulation of Wnt3a. In our opinion the paper has sufficient physiological meaning – it shows that metformin may improve regenerative potential of EMS-affected ASCs through acceleration of their proliferation and enhancement of viability.

Currently, we cannot provide the additional experiments showing metformin influence on differentiation process, though, as we mentioned in the first review, we have planned such experiments. We believe that the results presented in the current paper may be of interest to many readers of the journal.

Yours sincerely,

Agnieszka Śmieszek